# Mutational Assessment in *NKX2-5* and *ACTC1* Genes in Patients with Congenital Cardiac Septal Defect (CCSD) from Ethnic Kashmiri Population

**DOI:** 10.3390/ijerph19169884

**Published:** 2022-08-11

**Authors:** Nadeem Ul Nazeer, Mohammad Akbar Bhat, Bilal Rah, Gh Rasool Bhat, Shadil Ibrahim Wani, Adfar Yousuf, Abdul Majeed Dar, Dil Afroze

**Affiliations:** 1Department of CVTS, Sheri-Kashmir Institute of Medical Sciences, Srinagar 190011, India; 2Advanced Centre for Human Genetics, Sheri-Kashmir Institute of Medical Sciences, Srinagar 190011, India

**Keywords:** congenital heart defects, hotspot mutations, birth defects, ventricular septal defect, atrial septal defect, defects and coarctation of aorta

## Abstract

(1) Background globe. The etiology of CHDs is complex and involves both genetic and non-genetic factors. Although, significant progress has been made in deciphering the genetic components involved in CHDs, recent reports have revealed that mutations in Nk2 homeobox5 (*NKX2-5*) and actin alpha cardiac muscle1 (*ACTC1*) genes play a key role in CHDs such as atrial and ventricular septum defects. Therefore, the present study evaluates the role of key hotspot mutations in *NKX2-5* and *ACTC1* genes of congenital cardiac septal defect (CCSD) in ethnic Kashmiri population. (2) Methods: A total of 112 confirmed CHD patients were included in the current study, of which 30 patients were evaluated for mutational analysis for hotspot mutations of *NKX2-5* and *ACTC1* genes. The total genomic DNA was extracted from the samples (cardiac tissue/blood) and were subjected to amplification for *NKX2-5* (exon 1 and 2), and *ACTC1* (exon 2) genes by using PCR specific primers to analyze the hotspot mutations in respective exons. The amplified products obtained were sent to Macrogen Korea for sequencing by Sanger’s method. (3) Results: Our results confirmed that not a single mutation was found in either hotspot exon 1 and 2 of *NKX2-5* and exon 2 of *ACTC1* in the patients included in the current study. Interestingly, a novel synonymous nucleotide variation leading to G > C transversion (GCG > GCC) was found in exon 2 of *NKX2-5* gene of CCSD patient. (4) Conclusions: The current findings demonstrated the role of *NKX2-5* and *ACTC1* in cardiac development. The study will provide an insight in understanding the genetic etiology and highlights the role of newly identified mutations in patients with CDS’s in ethnic Kashmiri population. In silico findings revealed amino acid changes, splice site variation and the creation of new site. Furthermore, the study warrants complete screening of genes involved in CCSDs.

## 1. Introduction

Birth defects are defined as deformities of structure, function, or body metabolism. They have emerged as one of the major causes of morbidity and mortality worldwide [1]. 

Approximately, 8 million infants across the globe are born every year with serious birth defects. Birth defects affect around 33 in 1000 babies in the United States [2]. Owing to multifactorial etiologies, the majority of birth defects are difficult to identify before the onset of complete manifestations [3]. Moreover, significant progress has been achieved to validate the environmental factors associated with non-syndromic birth defects. However, understanding the genetic basis of birth associated defects is yet to be elucidated completely [4]. Among the birth defects, CHDs constitute a major proportion of clinically manifested birth defects and a key component of pediatric cardiovascular disease, with an estimated incidence of 6 to 9 per 1000 live births and around 40% of newborn babies die with CHDs in early infancy [5,6,7]. Epidemiologic and pathological investigations demonstrated that the etiology of CHDs is heterogeneous, which not only includes mutations in genes associated with cardiac genes, but also includes chromosomal anomalies, single gene disorders, epigenetic modifications, and genetic susceptibility to environmental exposures [4,8]. Aberrations in developmental pathways such as homeobox regulating genes viz *NKX2-5* and cardiac developmental genes such as *ACTC1* could trigger abnormal cardiac development [9]. Further, high throughput techniques targeting these genes and associated signaling pathways has led to the expansion of a multitude of mouse models with cardiac developing defects. These can provide a new platform for researchers to identify critical genes for normal cardiac development [10,11].

Various genetic aberrations are responsible for the etiology of CHD. Among them, *NKX2-5* plays a vital role in cardiac morphogenesis and acts as a transcriptional factor, regulating gene expression, which is essential in the early developmental stage and acts as a potential marker for cardiac myogenesis [12,13]. However, loss or mutation in one copy of the *NKX2-5* gene displayed a myriad of cardiac malformations that includes AV bundle scattering, reduction in HIS, altered expression of gap junction proteins, and variation in cellular density of AV node. This indicates that *NKX2-5* plays acritical role in heart development [14]. Additionally, in the recent past, the *NKX2-5* gene has been reported to play a key role in postnatal cardio protection [15]. The human *NKX2-5* gene maps to chromosome 5q34 and consists of two exons that encode a 324-amino acid protein [16]. Previous studies evaluated the exonic regions of the *NKX2-5* gene; however, recent findings instead investigate the mutations and associated CHDs in the intronic and promoter regions of the *NKX2-5* gene [17]. Functional analysis has elucidated the mechanism of action in various mutations of the *NKX2-5* gene and future studies are required in order to understand the involvement of *NKX2-5* in complex cardiac abnormalities [18].

*ACTC1*, a key member of conserved actin family, plays a critical role in cardiac muscle contraction and the physiology of the heart in association with troponins (C, I, and T) and tropomyosin [19]. The location of the *ACTC1* gene is on the q (long) arm of chromosome 15 at position 14 (15q14) [20]. *ACTC1* proteins are involved in cardiac muscle contraction and, thus, are essential and constitute a major part of the sarcomere thin filaments [21]. Lack of an *ACTC1* gene in mouse models exhibits myofibrillar disarray and embryonic lethality [22]. This plays a critical role in structural and functional integrity of cardiomyocytes, as only a few mutations in the *ACTC1* gene of humans have been reported. Recent evidence suggests that genetic defects in the *ACTC1* gene are associated with IDC (idiopathic dilated cardiomyopathy), FHC (familial hypertrophic cardiomyopathy), and ASD and that additional mutations in *ACTC1* gene might implicated in such type of cardiomyopathy [23].

Thus, with advancement in molecular genetics, genetic diagnosis provides more specific chromosomal analysis and copy number variants that emerge as critical factors for non-syndromic and syndromic CHDs [24,25]. Therefore, establishing precise diagnostics is of utmost importance, not only for clinical management and follow-up with patients in the future, but also to bring awareness to associated reproductive risks and future family planning. Playing a critical role in CHDs, the *NKX2-5* and *ACTC1* mutational analysis has not been reported in the Kashmiri population. Therefore, evaluating the role of genetic mutations such as the *NKX2-5* and *ACTC1* genes becomes important. The association with CHDs is important in patients of ethnic Kashmiri populations.

Therefore, the current study aimed to evaluate the mutational analysis of hotspot mutations of *NKX2-5* and *ACTC1* and their correlation with CHDs such as ASD patients of the ethnic Kashmiri population.

## 2. Materials and Methods

### 2.1. Study Design

The current study was designed and conducted in the Department of Cardiovascular Thoracic Surgery (CVTS) and Advanced Centre for Human Genetics, Sheri Kashmir Institute of Medical Sciences (SKIMS), Srinagar.

### 2.2. Cases

Only those subjects who qualified for the surgery were enrolled. However, the inclusion criteria for the patients included the operable cases subjected to surgery on cardiopulmonary bypass where tissue sample from the right atrium was possible. Exclusion criteria included advanced inoperable cases not fit for surgery, those on conservative management, and patients who did not give consent for inclusion for the current study. The subjects included were patients operated on for cardiac septal defects between August 2011 and December 2013. The patients with double chamber right ventricle (DCRV) and ventricular septal defect (VSD) underwent intracardiac repair with patch closure of VSD. Two patients with TA with ASD and VSD underwent bidirectional Glenn shunts while a repair of defect was performed in patients with an AV canal defect. A few patients who were operated on previously were also included. A written pre-informed consent was obtained from each patient. Demographic and clinic-pathological characteristics of each patient were recorded in a questionnaire. The study was approved by the ethical committee of the SKIMS.

### 2.3. Sample Collection/Storage

A total of 2–3 mL of peripheral blood was collected from each subject in a sterile 15 mL tube containing 10 µL/ml of 0.5 M EDTA (pH 8.0) as an anticoagulant and stored at −20 °C for future use. Approximately 500 mg of surgically resected cardiac tissues were collected directly into sterile vials and stored at −70 °C for molecular analysis. All the collected cardiac tissues and blood samples were used for mutational analysis of the *NKX2-5* and *ACTC1* genes.

### 2.4. Extraction of Genomic DNA and PCR Amplification

Genomic DNA was extracted from the above collected blood samples and cardiac tissues by kit method procured from Qiagen. To evaluate the role of some key genes in CHD, mutational profiles of the *NKX2-5* and *ACTC1* genes were performed by selecting 30 patients (20 cardiac tissue samples and 10 blood samples) from the current study who had clinical CCSDs. PCR amplification was performed on each extracted DNA samples with specific primers of the exons *NKX2-5* and *ACTC1* genes. The PCR amplification was performed using the forward primer, 5′-CGGCACCATGCAGGGAAG-3′ and reverse primer, 5′-AGGGTCCTTGGCTGGGTCGG-3′ for exon 1 of *NKX2-5* and forward primer, 5′-GCGCTCCGTAGGTCAAGC-3′ and reverse primer, 5′-TAGGGATTGAGGCCCACG-3′ for exon 2 of *NKX2-5* gene. In the case of *ACTC1* gene, the exon 2 was amplified using the forward primer, 5′-GATTATATTCCTGACATGGTGAGAG-3′ and reverse primer, 5′-GTAACTGTCCCCAGAGCCCA-3′. The PCR program conditions was 5 min at 95 °C, 35 s at 95 °C, 35 s at 60 °C for exon 1 and 2 of *NKX2-5* and 35 s at 55 °C for exon 2 of the *ACTC1* gene, 50 s at 72 °C, and 7 min at 72 °C.

### 2.5. DNA Sequencing

The crude PCR products of the samples were sent to Macrogen Korea for sequencing for further analysis.

### 2.6. Putative Role of the Variants

The bioinformatic analysis was performed using the gene mania software suite (Ontario Genomics Institute, Toronto, ON, Canada). This analysis helped to predict the interaction of the target gene(s) with other genes based on the physical, genetic, protein, and shared domain based on the color coding.

## 3. Results

A total of 112 confirmed CHD patients, enrolled in the Department of CVTS, SKIMS, Srinagar were recruited in the present study. Demographic data of 30 subjects which were selected for Sanger sequencing based on inclusion criteria revealed that about 88.12% of patients were from rural areas and 11.92% of patients were from the urban areas of Kashmir Valley of Northern India. As far as the gender distribution is concerned, 39.97% of the patients were males and 60.03% were females. The mean age of study subjects was 27.6 ± 12.8 years with the youngest subject being 5 years and the oldest being 53 years old. Most of the patients were in their first, second, third, and fourth decade of life. The majority of patients had ASD (40%), followed by VSD (20%). In addition to septal defects a few patients included in the study had other congenital cardiac abnormalities (Table 1). Out of 30 patients, 6 (20%) patients were the products of consanguineous marriages. However, two ASD patients had family history of same congenital cardiac disease. Moreover, three ASD patients (10%) had genetic syndromes (Down’s Syndrome), with recurrent chest infections (63.33%) and palpitations (80%) being the most frequent symptoms. The majority of the patients in the current study had common symptoms of breathlessness (66.33%) and fatigue (80%) followed by cough (56.66%) and poor weight gain (46.66%). Additionally, recurrent fever and feeding problems were present in 46.66% and 33.33% of patients, respectively. Three patients had a history of cyanotic spells (two Tetralogy of Fallot (TOF) patients and one Tricuspid Atria (TA) patient) whereas one patient with ASD had a history of convulsions.

The clinical presentations of CHD subjects in the current study are murmurs (80%), tachycardia (33.3%), crepitation (33.3%), hypertension (33.3%), clubbing (10%), cyanosis (10%), and rhonchi (10%). However, few patients had peripheral edema (13.33%), chest in drawing (6.66%), engorged neck vein (6.66%), and enlarged tender liver (6.66%). Fixed splitting of S2 was present in all ASD patients. The majority of patients (70%) were having NYHA (New York Heart Association) class II symptoms at presentation whereas (26.78%) of patients were in NYHA class III.

As for the operating mode of ASD patients,85.3% of patients were operated on with a beating heart, 14.6% were operated on with an arrested heart, 68.5% patients underwent patch closure (pericardium/PTFE patch) and 31.4% patients underwent primary closure. Patch closure of ASD with repair of PS was performed in three patients who had pulmonary stenosis (PS) in addition to ASD. However, patch closure with baffling of partial anomalous pulmonary venous connection (PAPVC) was performed in one patient. Additionally, patch closure of VSD was performed in all VSD patients. Intracardiac repair was performed in the TOF patient.

A post-operative complication such as atrial fibrillation (AF) was developed in four patients (13.33%) and ventricular tachycardia (VT) in one patient (3.33%). However, three patients (10%) developed pneumonia which was managed aggressively with antibiotics, and two patients (6.66%) developed wound infections which were managed by intravenous antibiotics and repeated aseptic dressings.

### Molecular Analysis of *NKX2-5* and *ACTC1* Genes

To identify any genetic defect of patients in the current study, such as having underlying CHD and associated defects, we performed a molecular analysis of the key genes *NKX2.5* and *ACTC1*, which play a critical role in both structural and functional development of cardiomyocytes. To perform molecular analysis of the *NKX2-5* and *ACTC1* genes, genomic DNA isolated from the samples (blood and tissues) were subjected to PCR analysis. PCR amplified product of exon 1 of *NKX2-5* product was 405 bp. However, PCR amplified product of exon 2 of *NKX2-5* was 470 bp. While as PCR amplified product of exon 2 of *ACTC1* was 405 bp, respectively.

The products obtained from 30 CCSD DNA samples after PCR amplification were analyzed for any mutation in *NKX2-5* (exon 1 and 2) and *ACTC1* gene by direct sequencing. The sequencing data were analyzed and interpreted manually or by using ClustalW2 software (EML-EBI, Welcome Genome Campus, Hinxton, UK). Although, we could not find any mutation in exon 1 and 2 of *NKX2-5* gene as well as exon 1 of *ACTC1* in the subjects included in the study as shown in Figure 1a,b and Figure 2a,b, respectively.

However, a novel nucleotide variation leading to G > C transversion was observed in exon 2 in one of the blood samples of CCSD patient. The change lies in the coding region of exon 2, which changes codon GCG > GCC (Figure 3).

To determine the effect of this change on the functioning of *NKX2-5* gene, the bioinformatics in silico analysis of this variation was performed by ESE (v.3.0). It showed that the variation creates new sites that could influence the physiology of the *NKX2.5* gene as shown in Figure 4. Moreover, the splicing effect of the variant needs to be confirmed by in vitro analysis. Furthermore, the in silico analysis using the mutation taster platform revealed that the variation leads to the amino acid change (A119P) and influences the regulatory elements such as DNase1, Open Chromatin, DNase1 Hypersensitive Site H3K36me3, Histone, Histone 3 Lysine 36 Tri-Methylation H3K27me3, Histone, and Histone 3 Lysine 27 Tri-Methylation DNA hypersensitive regions.

## 4. Discussion

CHD is among the most prevalent and fatal of all birth defects. Therefore, a significant number of research studies in the past and present aim to elucidate the underlying factors involved in CHD. The accurate etiology of this complex disease is still ambiguous [26]. Deciphering its causes, however, is complicated, as many patients affected by CHD have no family history of the disease. There is also widespread heterogeneity of cardiac malformations within affected individuals. Nonetheless, there have been tremendous efforts towards a better understanding of the molecular and cellular events leading to CHD [10]. Notably, certain cardiac-specific transcription factors have been implicated in mammalian heart development and disruption of their activity has been demonstrated in CHD [18]. The homeodomain transcription factor *NKX2-5* is an important member of this group. Moreover, *ACTC1s* are the predominant component of the sarcomeric thin filaments and are essentially required for cardiac muscle contraction [27]. Mutations in this gene have been shown to be associated with CHDs including ASD [28]. Given the importance of *NKX2-5* and *ACTC1* in the physiological and functional integrity of cardiomyocytes and associated defects such as ASD in patients of CHDs, the genetic and molecular analysis is of critical importance. In the current study, we analyzed the mutational profile of *NKX2-5* gene and exon 2 of *ACTC1* gene in CCSD patients. The interaction of *NKX2-5* and *ACTC1* with other critical genes will help in understanding the pathogenesis of the disease in a precise way as shown in Figure 5.

The *NKX2-5* gene belongs to the NK-2 family of homeodomain-containing transcriptional factors, which are conserved in many organisms ranging from *Drosophila melanogaster* to humans [29,30]. Commonly known as the ‘tinman’ gene in *Drosophila melanogaster*, it is expressed in the growing dorsal vessel and is believed to be equivalent to the vertebrate heart [31]. The tinman gene mutation causes a malformation of the heart in the embryo, suggesting that tinman is indispensable for heart formation in Drosophila [32]. Recent evidence suggests that humans and murine models during the early stage of cardiac development have abundant expression of the *NKX2-5* gene [33]. Intriguingly, *NKX2-5* knockout murine models showed serious early embryonic lethality [34]. In the present study, both the exons of *NKX2-5* gene were screened for mutational profile in 30 CCSD patients. No mutation was detected in exon 1 of this gene. Indeed, more than 40 heterozygous *NKX2-5* germline mutations have been observed in individuals with CHD that are spread along the coding region, with many shown to impact protein function. *NKX2-5* appears to be hypermutable, yet the overall detection frequency in sporadic CHD is about 2% and *NKX2-5* mutations are one-time detections with single positives or private to families. Although family studies have been useful in uncovering genes linked to CHD, population-based studies showed rare detection of mutations in unrelated patients with CHD [35]; therefore, this indicates a multifactorial cause of disease. These findings explain the lack of mutations in the *NKX2-5* gene in our study group. Moreover, these earlier studies found familial and private mutations in patients, and mutations were rare in sporadic cases. However, our genetic analysis was based on DNA from diseased cardiac tissues of unrelated patients. An interesting finding of this study was the detection of a novel heterozygous nucleotide change (G > C) observed for the first time in the exon 2 of *NKX2-5* gene in one of the 30 samples (3.3%). This causes substitution of codon GCG > GCC coding for amino acid alanine. The in silico analysis showed that this variation creates new sites that could influence the physiology of the *NKX2-5* gene and have an effect on the regulatory sites. Such variations eventually serve as risk factors for CHD, as shown in Figure 6.

The present study also examined the mutational profile of the coding exon 2 of the *ACTC1* gene in 30 CCSD patients from Kashmir valley. No mutation was found in exon 2 of the *ACTC1* gene in our study. The lack of mutations of the *ACTC1* gene as found in our study is in discordance with the other studies where patients with contiguous gene syndromes and chromosome 15q deletions spanning the *ACTC1* gene were reported previously [36]. Several of these patients present with CHD including ASD, suggesting a gene locus of importance for cardiac septal formation on chromosome 15q. Furthermore, a recent study by Monserrat et al. [36] identified septal defects in some individuals with cardiomyopathy and *ACTC1* mutations. Interestingly, Matsson et al. [35] identified dominant *ACTC1* gene mutations in patients with isolated ASD. These mutations were associated with a marked clinical variability from an asymptomatic shunt to a severe cardiac decompensation. Moreover, they also identified a case with ASD and a 17 bp deletion in exon 2 of *ACTC1* gene. This inconsistency may be because of the studied populations belong to different geographical regions or ethnicities and due to the small number of patients included in our study.

## 5. Conclusions

Moreover, the completion of the Human Genome Project, the advances in genetic technology, and single gene defects explain the syndromes associated with CHDs. CCSDs defects constitute a major proportion of clinically significant birth defects and are an important component of pediatric cardiovascular disease. Mutations in *NKX2-5* proteins result in a myriad of CHDs, which are mostly either atrio-ventricular conduction defects or septal defects such as ASD, VSD, hypoplastic left heart syndrome (HLHs), double outlet right ventricle (DORV), transposition of great arteries (TGA), interrupted aortic arch (IAA), cardiac outflow tract (OFT) defects, and coarctation of aorta (CoA). Previous studies revealed that mutations in *NKX2-5* and *ACTC1* genes lead to isolated cardiac septal defects. However, the current study concludes that though important for cardiac development, mutations in *NKX2-5* and *ACTC1* genes are rare in patients with CCSD. This is the first study to evaluate the role of *NKX2-5* and *ACTC1* gene mutations in CCSD patients from the Kashmiri population, which will involve further counselling for patients and study of genetic variations on an individual patient basis. The small sample size targeting a genetically heterogeneous population is the main limitation of this study.

## Figures and Tables

**Figure 1 ijerph-19-09884-f001:**
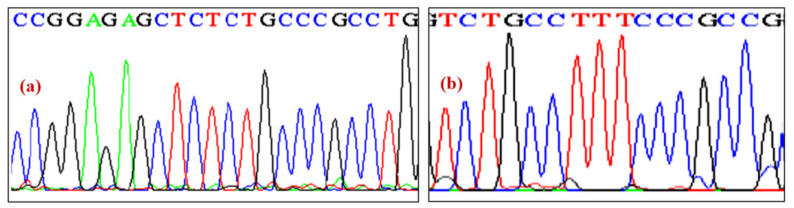
(**a**,**b**) The chromatogram of exon 1 and 2 of *NKX2-5* gene.

**Figure 2 ijerph-19-09884-f002:**
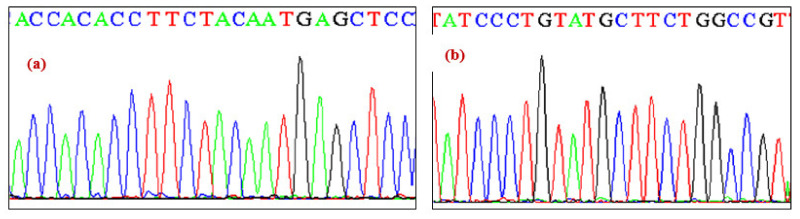
(**a**,**b**) The chromatogram of exon 1 of *ACTC1* gene.

**Figure 3 ijerph-19-09884-f003:**
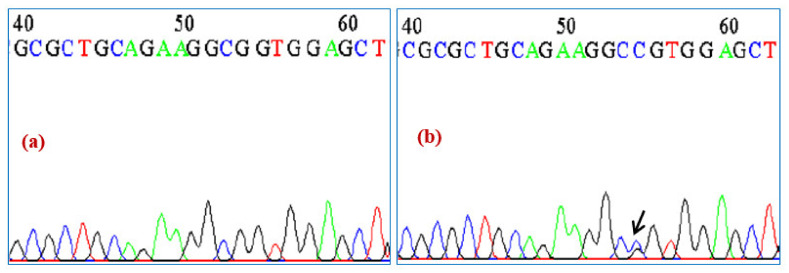
(**a**,**b**) The chromatogram of exon 2 of *NKX2-5* gene showing sequence variation GCG > GCC. (**a**) Wild type sequence. (**b**) Mutant sequence.

**Figure 4 ijerph-19-09884-f004:**
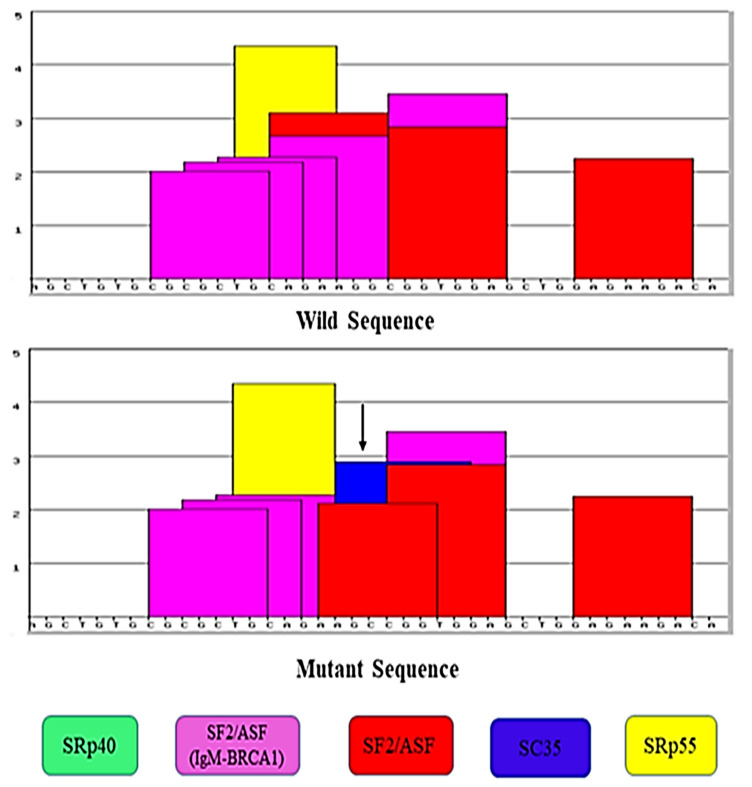
Effect of genetic variation on the Exonic Splicing Enhancers (ESEs) according to ESE prediction tool. ESE finder enables to recognize the potential ESE sites. The elevation of the colored bars represents the motif scores, and the girth of the bars indicates the length of the motif. Bars in red, yellow, blue, purple, and green indicate potential binding sites for serine-arginine (SR) proteins SF2/ASF, SRp55, SC35, SF2/ASF (IgM-BRCA1), and SRp40, respectively. Upper Panel signifies the ESE analysis of normal sequence and lower panel denotes the ESE sequence with effect on splice sites. From Figure 4, we can predict that there is a change in the potential ESE sites as can be seen from change in the bars (change in the potential splicing sites) that might increase the disease susceptibility.

**Figure 5 ijerph-19-09884-f005:**
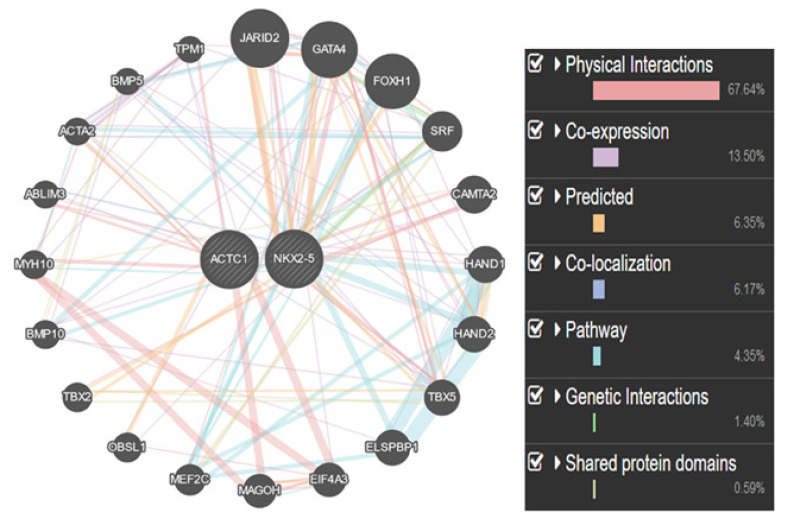
The interaction of *NKX2-5* and *ACTC1* with other critical genes using the gene mania software suite.

**Figure 6 ijerph-19-09884-f006:**
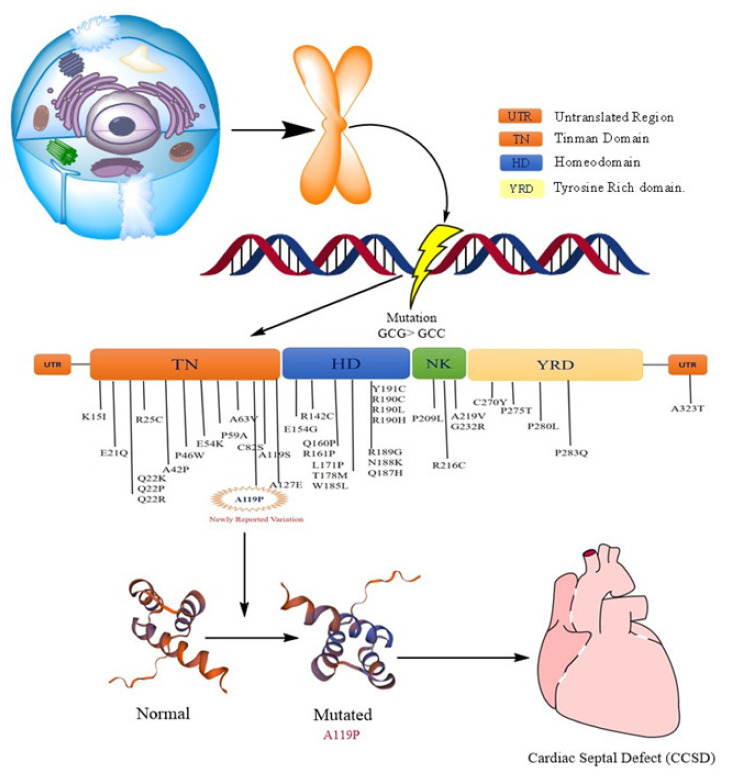
Schematic representation of the effect of the newly identified variant of *NKX2-5* and its effect on CHD.

**Table 1 ijerph-19-09884-t001:** Types and sex distribution of different congenital heart disease (n = 30).

**Types**	**Male (Number and % Age)**	**Female (Number and % Age)**
ASD	(n = 03, 10%)	(n = 09, 30%)
ASD with PAPVC	Nil	(n = 01, 3.33%)
ASD with VSD	(n = 01, 3.33%)	Nil
VSD	(n = 02, 6.66%)	(n = 04, 13.33%)
VSD with RSOV	(n = 01, 3.33%)	Nil
TOF	(n = 01, 3.33%)	(n = 01, 3.33%)
TA with ASD and VSD	(n = 02, 6.66%)	(n = 01, 3.33%)
DCRV with VSD	(n = 02, 6.66%)	(n = 02, 6.66%)
Gender distribution	39.97%	60.03%
**Age Distribution of Patients**		
**Age**	**Male (Number and % Age)**	**Female (Number and % Age)**
0–10 years	(n = 04, 13.3%)	(n = 03, 10%)
11–20 years	(n = 03, 10%)	(n = 04, 13.33%)
21–30 years	(n = 03, 10%)	(n = 04, 13.3%)
31–40 years	(n = 02, 6.66%)	(n = 03, 10%)
41–50 years	(n = 01, 3.33%)	(n = 02, 6.66%)
51–60 years	(n = 01, 3.33%)	Nil
**Consanguinity, Disease Pathologies, Other Clinical Symptoms of Patients**		
Consanguineous marriage	(n = 04, 13.33%)	(n = 02, 6.66%)
Recurrent chest infection	(n = 6, 20%)	(n = 13, 43.33%)
Palpitation	(n = 9, 30%	(n = 15, 50%)
Breathlessness	(n = 7, 23.33%)	(n = 12, 40%)
Fatigue	(n = 11, 36.66%)	(n = 13, 43.33%)
Cough	(n = 7, 23.33%)	(n = 10, 33.33%)
Poor weight gain	(n = 7, 23.33%)	(n = 7, 23.33%)
Recurrent fever	(n = 6, 20%)	(n = 8, 26.66%)
Feeding problems	(n = 4, 13.33%)	(n = 9, 30%)

**Abbreviations:** ASD; atrial septal defect, PAPVC; partial anomalous pulmonary vein connection, VSD; ventricular septal defect, RSOV; ruptured sinus of Valsalva aneurysm, TOF; Tetralogy of Fallot, TA; tricuspid atresia, DCRV; double-chambered right ventricle.

## Data Availability

The data accumulated from the current work by the Advanced Centre for Human Genetics, Sher-i-Kashmir, Institute of Medical Sciences, Srinagar, J&K, India, can be shared with scientists on application to the corresponding or principal investigators of this study.

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
