# Peer review of "Mutational Assessment in NKX2-5 and ACTC1 Genes in Patients with Congenital Cardiac Septal Defect (CCSD) from Ethnic Kashmiri Population"

_ijerph, 2022, doi:10.3390/ijerph19169884_

Round 1

Reviewer 1 Report

The topic dealt with in this article is of fundamental importance for the understanding of the mechanisms underlying important congenital and non-congenital pathologies that affect a large number of people today. Unfortunately, the methods used are merely reductive and simple, thus demonstrating the presence of a mutation, but which still fails to be confirmed in basic research, much less in clinical practice.

The basic idea and structure of the article as a whole are well done, but the lack of further evidence makes the article less resonant on a scientific level.

To better appreciate the discovery, it is advisable to understand or at least try to understand what the mechanisms really affected by the genetic modification may be, using laboratory animals on which it is possible to test in vivo what the effects of this mutation are and possibly evaluate whether the recovery of the normal genetic sequence leads to a situation of normality, or if there are mechanisms that circumvent the problems it causes.

Reviewer 2 Report

In their paper “Evaluation of newly identified mutation of NKX2-5 and role of ACTC1 in congenital cardiac septal defect (CCSD) patients in ethnic Kashmiri population”, the authors evaluate, for the first time, the mutations of NKX2-5 and ACTC1 genes in the Kashmiri population. Despite the novelty, I believe that the manuscript needs substantial reorganization and implementation. I ask the authors for several major revisions and additional details.

  1. Since both NKX2-5 and ACTC1 genes are assessed in this study, consider changing the title, for example: “Mutation assessment in NKX2-5 and ACTC1 genes in patients with congenital cardiac septal defect (CCSD) from ethnic Kashmiri population”
  2. Please check the abbreviations and remove those used only once.
  3. Introduction:
    1. The authors use the terms CHD, ASD, and CCSD interchangeably throughout the paper creating confusion for the reader. Since the population of the study includes ASD and VSD, the introduction should provide information about these pathologies rather than general information about CHD (lines 1-61). Please rewrite the introduction accordingly.
    2. I suggest shortening the introduction to make it easier to read. Some parts are off-topic, such as lines 63-71, 76-82, 96-98 (only the coding region is evaluated there), and 110-111, thus consider removing these lines.
    3. Some parts seem more suitable for other paragraphs. Specifically, consider moving lines 72-73 and 98-101 to the conclusion; lines 82-90 and 125-127 to the discussion.
  4. Materials and methods
    1. The number of patients enrolled is not clear: 112 at line 171, 107 at line 180. Please clarify the number in paragraph 2.2
    2. What are the exact exclusion criteria that lead to the screening of only 30 patients? Please describe them in paragraph 2.2
    3. Indicate the exact number of samples: move lines 218-221 to paragraph 2.3.
    4. Why did the authors perform analyses on 20 cardiac tissue and 10 blood samples?
    5. Line 155: How was the DNA extraction performed?
    6. Lines 227-230 are suitable for paragraph 2.4, line 163.
    7. Please include the detail of bioinformatics analysis in materials and methods.
  5. Results
    1. After specifying the number of patients selected for the analysis, the characteristics of the final cohort should be presented in a table for more readability and the results need to be commented on accordingly. Therefore, the results should be rewritten to describe the actual cohort.
    2. Is there familiarity in the selected cohort or not? Lines 180-181 and 289 appear contrasting.
    3. Table 1 needs a caption with abbreviations or please consider using the extended-expression.
    4. Lines 225-226 are not needed.
    5. Sentence 234-236 contrast with lines 249-250. Moreover, please indicate the gene NKX2-5 (line 250).
    6. Lines 255-256 belong to the discussion.
  6. Discussion
    1. Lines 270-283 are redundant with other parts of the manuscript. Consider removing.
    2. The discussion is not clear and should be rewritten based on results.
    3. The consequences of the changes described in figure 5 should be reported and commented on.
    4. Figure 1 belongs to the discussion as it describes the importance of NKX2-5 and ACTC1 genes in the pathogenesis of the disease.
    5. Figure 6 should be improved: some mutations are not legible. Moreover, the figure needs a caption with abbreviations.
  7. A paragraph reporting the limitations of the study is missing.

Round 2

Reviewer 2 Report

Dear authors,

Thanks for your comments. I have read your manuscript and still have few observations.

1. I have founded some abbreviations that have not been introduced properly, for example VSD used on page 4 line 14 was first introduced on page 5 line 21, CCSD used on page 2 line 2 was first introduced on page 13 line 3. Some abbreviations on page 13 between lines 4 and 8 are only used here, so remove these abbreviations. Please check manuscript for all the abbreviations.

2. page 13 line 13: Please change “only limitations” with “main limitation”.

3. page 4 line 27: were the 30 patients randomly selected?

4. As previously indicated in comment 5, the description of the population seems confounding. Since the analyses were performed on the subgroup of 30 patients, the manuscript should focus on this subgroup rather than the 112 individuals. Therefore, please rewrite methods and results accordingly.

5. I’m sorry but table 1 still needs some changes, it should presents data for the 30 individuals including  mean age, gender, pathologies, symptoms. I have indicated some suggestions as comments in the pdf file.

Author Response

Reviewer 2 comments:

Thanks for your comments. I have read your manuscript and still have few observations.

Comment 1: I have founded some abbreviations that have not been introduced properly, for example VSD used on page 4 line 14 was first introduced on page 5 line 21, CCSD used on page 2 line 2 was first introduced on page 13 line 3. Some abbreviations on page 13 between lines 4 and 8 are only used here, so remove these abbreviations. Please check manuscript for all the abbreviations.

Reply to comment 1: As suggested by the reviewer, we have incorporate abbreviations at right places and removed repetitions in entire manuscript.

Comment 2: page 13 line 13: Please change “only limitations” with “main limitation”.

Reply to comment 2: As suggested by the reviewer, we have replaced ‘‘only limitations’’ with main ‘‘limitation’’

Comment 3:  page 4 line 27: were the 30 patients randomly selected?

Reply to comment 3: The 30 subjects were selected on the basis of inclusion criteria mentioned in materials and methods section under subheading cases of the manuscript.

Comment 4: As previously indicated in comment 5, the description of the population seems confounding. Since the analyses were performed on the subgroup of 30 patients, the manuscript should focus on this subgroup rather than the 112 individuals. Therefore, please rewrite methods and results accordingly.

Reply to comment 4: We have performed the molecular analysis on 112 patients, but based on the inclusion criteria and the severity of the patients we further shortlisted 30 patients for the Sanger sequencing in order to ascertain the role of exon 1 and 2 of NKX2-5 and exon 2 of ACTC1 gene. The results of the same are mentioned in the result section of the manuscript.

Comment 5: I’m sorry but table 1 still needs some changes, it should presents data for the 30 individuals including mean age, gender, pathologies, symptoms. I have indicated some suggestions as comments in the pdf file.

Reply to comment 5: As suggested by the reviewer, we have edited the said table and included the data of 30 patients which includes pathologies as well as symptoms in the table 1 of the manuscript.
